# Relationship of Zonulin with Serum PCSK9 Levels after a High Fat Load in a Population of Obese Subjects

**DOI:** 10.3390/biom10050748

**Published:** 2020-05-11

**Authors:** María Molina-Vega, Daniel Castellano-Castillo, Lidia Sánchez-Alcoholado, Isaac Plaza-Andrade, Gabriel Perera-Martin, Amanda Cabrera-Mulero, Jose Carlos Fernández-García, Bruno Ramos-Molina, Fernando Cardona, Francisco J. Tinahones

**Affiliations:** 1Department of Endocrinology and Nutrition, Virgen de la Victoria University Hospital, 29010 Malaga, Spain; molinavegamaria@gmail.com (M.M.-V.); josecarlosfdezgarcia@hotmail.com (J.C.F.-G.); fjtinahones@hotmail.com (F.J.T.); 2Instituto de Investigación Biomédica de Málaga, Universidad de Málaga, 29010 Malaga, Spain; danie__cc@hotmail.com (D.C.C.); l.s.alcoholado@gmail.com (L.S.-A.); isaacplazaandrade@gmail.com (I.P.-A.); gabriel.perera.martin@gmail.com (G.P.-M.); amanda.c.mulero@hotmail.com (A.C.-M.); 3Centro de Investigación Biomédica en Red de la Fisiopatología de la Obesidad y Nutrición, Instituto de Salud Carlos III, 28029 Madrid, Spain; 4Biomedical Research Institute of Murcia (IMIB-Arrixaca), 30100 Murcia, Spain; brunoramosmolina@gmail.com

**Keywords:** PCSK9, high fat load, intestinal permeability, zonulin, apolipoproteins

## Abstract

Despite the fact that circulating levels of proprotein convertase subtilisin/kexin type 9 (PCSK9) remain unchanged after fat load in healthy lean individuals, PCSK9 has been suggested to have a role in postprandial lipemia regulation in obese individuals. On the other hand, intestinal permeability and endotoxemia have been observed to increase more in obese individuals than in non-obese individuals after a lipid load. This study aimed to analyze the relationship between PCSK9, intestinal permeability, and endotoxemia after a high fat load in obese individuals. We included 39 individuals with morbid obesity. Serum PCSK9 levels, intestinal permeability marker (zonulin), endotoxemia markers (LPS and LBP), and lipid parameters were measured before and after 3 h of fat load. A significant rise in triglycerides, apolipoprotein A1, zonulin, LPS, and LBP, and a significant decline in PCSK9, were observed after a lipid load. Linear regression analysis showed that low-density lipoprotein cholesterol (LDL-C) was independently related to PCSK9 at baseline, whereas both zonulin and LDL-C were independently related to PCSK9 levels after fat load. A relationship between zonulin and PCSK9 levels after fat load in individuals with morbid obesity may exist.

## 1. Introduction

Increasingly, in developed countries, daily excessive food intake causes a permanent postprandial status. Postprandial lipemia is characterized by an increase in triglyceride-rich lipoproteins (TRLs) containing apolipoprotein B-48 (apoB-48) produced by the enterocyte [1], a decrease in high-density lipoprotein cholesterol (HDL-C), and stable concentrations of low-density lipoprotein cholesterol (LDL-C) [2]. The equilibrium between the secretion and clearance of these TRLs is altered in obesity and diabetes and is influenced by environmental factors, resulting in the accumulation of remnant lipoproteins (RLPs) that have been associated with a higher risk of cardiovascular disease [3].

Proprotein convertase subtilisin/kexin type 9 (PCSK9) is a member of the proprotein convertase family that signals the degradation of LDL-C receptor (LDL-R) via lysosome action, which may reduce LDL-C clearance [4]. It was found that individuals with *PCSK9*- LOF (loss of function) variants had lower levels of LDL-C; therefore, PCSK9 inhibition was proposed as a possible therapeutic approach to hypercholesterolemia. Drug-mediated inhibition of PCSK-9 has been shown to markedly reduce LDL-C levels and cardiovascular risk disease [5]. In addition to its effect on LDL-C, another possible role of PCSK9 in TRLs metabolism has been proposed. However, the exact molecular mechanisms and the extent of postprandial lipid metabolism regulation by PCSK9 are still under study [6]. In vitro studies using Caco-2 enterocytes and studies in rodents have revealed a relationship between PCSK9 expression and apoB-48 secretion [6,7]. In humans, Cariou et al. [8] did not find any variation in PCSK9 levels after a fat load in healthy individuals, while Chan et al. [9] concluded that catabolism of triglycerides and apoB-48-containing chylomicrons may be regulated by PCSK-9 in individuals with obesity. Ooi et al. [10] observed that individuals with *PCSK9*-LOF had lower postprandial triglyceridemia, apoB-48, and total apoB levels, consistent with the results observed in vitro and in rodents. On the other hand, it has been observed that PCSK9 inhibition is implicated in the clearance of pathogen lipids, such as lipopolysaccharide (LPS), through LDL receptor (LDLR) [11,12] while Grin et al. [13] reported that high concentrations of PCSK9 directly suppress uptake of lipopolysaccharide by hepatocytes.

In recent years, there has been an increased focus on the effect of gut microbiota dysbiosis on metabolic diseases. High-fat diets (HFD) have been associated with changes in gut microbiota, affecting intestinal membrane integrity, which in turn increases intestinal permeability and endotoxemia [14]. However, these effects differ between normal weight and obese individuals, with obese individuals having higher intestinal permeability and endotoxemia compared with non-obese controls after a lipid load [15,16].

In view of these findings, we hypothesized that the effect of a high fat load on intestinal permeability and endotoxemia could be associated with circulating PCSK9 levels in obese individuals. Thus, the aim of this study was to analyze the effect of a fat load on intestinal permeability, endotoxemia, and PCSK9 levels, as well as their possible association, in a population of obese individuals.

## 2. Material and Methods

### 2.1. Patients

For this study, we recruited 39 individuals aged over 18 years with morbid obesity (body mass index, BMI ≥ 40 kg/m^2^) awaiting bariatric surgery at Virgen de la Victoria University Hospital, Málaga. Exclusion criteria included the presence of diabetes mellitus (DM), cardiovascular disease, arthritis, acute inflammatory disease, infectious disease, renal disease, or patients receiving drugs that could alter the lipid profile at the time of study inclusion.

### 2.2. Study Protocol

All study participants received nutritional recommendations to follow a similar diet regarding carbohydrate, protein, and lipid contents on the day before the clinical evaluation, and were instructed to fast for at least 12 h before sample collection. Blood samples were collected from the antecubital vein at baseline and 3 h after the fat load and placed in vacutainer tubes (BD Vacutainer™). Fat load was performed using a preparation of 100 mL containing 50 g fat that comprised 30% saturated, 49% monounsaturated, and 21% polyunsaturated fatty acids, with less than 1 g lauric acid, less than 1 g myristic acid, 4.8 g palmitic acid, 1.4 g stearic acid, 27.7 g oleic acid, 9.6 g linoleic acid, 1.4 g behenic acid, and 0.5 g lignoceric acid (patent N° P201030776) [17]. Participants were allowed to drink only water during the test and no physical exercise was permitted. The samples were immediately stored at −80 °C until analysis.

Through a structured interview of the participants, we obtained clinical data including age, medical history, current diseases, and associated treatment. Anthropometrical data were also collected and included weight, height, and waist circumference (WC).

Serum PCSK9 levels were measured by ELISA (Human PCSK9 SimpleStep ELISA kit, Abcam, Canada). The inter-assay coefficient of variation (CV) was 4.6% and the intra-assay CV was 4.4%. Serum levels of zonulin were determined by ELISA (human zonulin ELISA kit by MyBioSource, catalog number: MBS749365). Both inter-assay and intra-assay CV were <10%. Triglycerides, total cholesterol (TC), and HDL-C were measured using standard enzymatic methods (Dimension Vista^®^, Siemens, Healthcare Diagnostics Inc. Newark, DE 19714, U.S.A.). LDL-C was calculated using Friedewald equation [18]. Insulin levels were measured by immunoassay using an ADVIA Centaur autoanalyzer (Siemens Healthineers, Erlangen, Germany). Insulin resistance was determined using the homeostasis model assessment of insulin resistance index (HOMA-IR), as described by Matthews et al. [19]. Apolipoprotein A1 (ApoA1), total apolipoprotein B (ApoB) and apolipoprotein CIII (ApoCIII) were measured with ELISA (Human Apolipoprotein A1 ELISA Kit by Thermo Scientific, Apolipoprotein B Human SimpleStep ELISA Kit by ABCAM and Human Apolipoprotein C3 ELISA Kit by Thermo Scientific respectively). As endotoxemia markers, LPS and LBP (lipopolysaccharide-binding protein) were measured with ELISA (Limulus Amebocyte Lysate Chromogenic Endpoint Assay kit; CV intra and inter-assay was 1.9% for LAL and LPB for a wide variety of species; CV intra-assay 6.6% and inter-assay 7.9% for LBP boths by HycultBiotech).

### 2.3. Ethics

This study was reviewed and approved by the local Ethics Committee of the Virgen de la Victoria University Hospital and was conducted according to the principles of the Declaration of Helsinki. The participants (all volunteers) provided signed consent after being fully informed of the study goal and its characteristics.

### 2.4. Statistical Analysis

Normal distribution of the variables was evaluated using the Kolmogorov–Smirnov test; data were expressed as mean ± SD (standard deviation). The hypothesis testing for continuous variables was performed using a t-test. Associations between the qualitative characteristics were tested using the χ2 test. The relationship between continuous variables was examined using correlation analyses (Pearson). Multivariate linear regression models were constructed, taking into account multicollinearity (through the variance inflation factor), using as dependent variables PCSK9 levels before and after fat load. The criterion used for selecting the best model was based on the Akaike information criterion. Values were considered to be statistically significant when *p* < 0.05. The analyses were performed using SPSS Statistics (version 25 for Windows; IBM Corp., Armonk. New York).

## 3. Results

Data from 39 subjects, 13 male and 26 female, were analyzed. Baseline characteristics of the population and the comparison of the lipid parameters, PCSK9 levels, intestinal permeability, and endotoxemia at baseline and 3 h after high fat load is presented in Table 1. A significant increase in triglycerides, Apo-AI, zonulin LPS activity, and LBP levels, and a significant decrease in serum PCSK9 levels were observed after a high fat load.

Correlation analysis (Table 2) showed positive correlations between serum PCSK9 levels and TC, LDL-C, and HDL-C at baseline and 3 h after fat load. 

No correlation was found between PCSK9 and triglycerides, apolipoproteins, LPS, or LBP at neither baseline nor 3 h post fat load. In addition, a positive correlation was observed between PCSK9, LDL-C, and zonulin after 3 h of fat load. However, we did not observe any correlation between PCSK-9 and zonulin at baseline. In addition, we found a significant correlation between LPS and Apo-AI at baseline (r = 0.340; *p =* 0.037) and 3 h after fat load (r = 0.390; *p* = 0.019).

In the multiple linear regression analysis, using PCSK9 at baseline as the dependent variable, we found a statistically significant independent relationship only with LDL-C levels (Table 3).

However, we found that both LDL-C and zonulin were factors independently related to PCSK-9 levels after 3 h of fat load (Table 4).

## 4. Discussion

Our results show that PCSK9 levels decrease after a fat load while zonulin and endotoxemia increase in individuals with morbid obesity. In addition, we found an independent association between PCSK9 and zonulin. On the other hand, we did not observe any relationship between PCSK9 and triglycerides.

Not many studies have analyzed the changes in PCSK9 levels after an oral fat load in humans. Cariou et al. [8] did not observe any changes in PCSK9 in a small sample of 10 healthy men. Conwey et al. [20] also found no differences in PCSK9 levels in a population of 34 healthy men and women after 4 weeks of a high-fat diet consisting of buttermilk. However, consistent with our observation, Ooi et al. [10] described a notable decrease in PCSK9 levels in a sample of 23 overweight (BMI 28.4 ± 3.2 kg/m^2^) men and women 4 h after fat load, suggesting that the response of PCSK9 may happen only in individuals with obesity.

As widely reported and recently reviewed by Rohr et al. [21], dietary fats directly affect the intestinal barrier by altering tight junctions, inducing oxidative stress and apoptosis of intestinal epithelial cells, which are prone to proinflammatory and barrier-disrupting cytokines, thus negatively modulating the composition of intestinal mucus and stimulating the enrichment of gut microbiota with barrier-disrupting species. These changes increase intestinal permeability and LPS translocation. Nevertheless, in both healthy lean [22,23,24] and obese individuals [25,26], the rise in LPS levels after high fat consumption has been reported. Vors et al., when comparing lean and obese individuals in a randomized, controlled, cross-over study, found that only the obese individuals had higher postprandial endotoxemia after fat load [16]. More recently, Genser et al. [15] showed that intestinal permeability after a lipid load among obese patients was twice as high as that among non-obese individuals. In addition, they concluded that there are subtle intestinal barrier alterations in the fasting state in obese individuals, which are clearly revealed by the challenge involved in the fat load. These findings agree with the results of our study and could explain the neutral effect of fat load on PCSK9 in healthy lean individuals [8,20].

Despite the fact that the relationship between PCSK9 and intestinal permeability has not been studied in depth, it is known that LDL-R is implicated in LPS clearance [11]. Moreover, previous studies have shown that the inhibition of PCSK9 and the consequent promotion of LDL-R favor LPS detoxification [27,28,29], indicating PCSK9 as a possible regulator of LPS clearance. Therefore, PCSK9 reduction after fat load could help in the clearance of higher plasma concentrations of LPS resulting from the increased intestinal permeability due to high fat consumption in obese individuals. We have found intestinal permeability to be independently related to PCSK9 after fat load, but we failed to find any relationship with LPS. Although endotoxemia is currently admitted to reflect intestinal permeability, postprandial LPS can also result from transcellular secretion associated with chylomicrons, so postprandial endotoxemia is not necessarily a reflection of intestinal permeability and the relationship between lipids and the intestinal barrier has not been elucidated [15].

Finally, we did not find any relationship between PCSK9 levels and TG, neither at baseline nor 3 h after fat load. Some researchers have reported a positive association between plasma PCSK9 and plasma TG in the general population and in individuals with different pathologies such as type 1 and type 2 DM, HIV, chronic kidney disease, or type III hyperlipidemia [6]. In addition, Chan et al. [9] found a positive association between PCSK9 levels and IDLs (intermediate-density lipoproteins) and an inverse relationship between PCSK9 levels and the fractional catabolic rate in a postprandial stable-isotope study involving a population of individuals with obesity. However, two studies analyzing the association between PCSK9 and lipoprotein kinetics obtained opposite results. Sullivan et al. [30], studying 39 nondiabetic obese individuals, did not find any relationship between PCKS9 levels and VLDL-TG secretion, VLDL-TG clearance, or plasma VLDL-TG levels, while Vèrges et al. [31] did not observe any association between PCSK9 and VLDL production rates in 38 individuals (with and without type 2 DM). As concluded by Dijk et al. [6], the available literature shows that the impact of PCKS9 on TRL metabolism is modest.

The main limitation of our study is the small sample size, although it is similar to other kinetic studies in humans. This study was designed to confirm an exploratory hypothesis so a control group was not included. In the future, we will most likely prepare a new study with a control group. An important strength is that, to our knowledge, our study is the first to report changes in PCKS9 levels after a fat load in morbidly obese individuals, and the possible relationship between PCSK9 and intestinal permeability. However, we have to take into account the limitation of the human zonulin ELISA kit used, which measures not only zonulin but also other proteins in the same family, such as haptoglobin. Therefore, in future studies, it will be necessary to use other markers of intestinal permeability, together with zonulin, to strengthen this relationship.

In conclusion, PCSK9 levels decrease while intestinal permeability and endotoxemia increase after a fat load in individuals with morbid obesity. There seems to be a relationship between PCSK9 and intestinal permeability in this context. Further studies are required to determine if our findings are reproducible and to understand the underlying mechanism implicated in the relationship between PCSK9, intestinal permeability, and lipoprotein metabolism in obese individuals.

## Figures and Tables

**Table 1 biomolecules-10-00748-t001:** Comparison between baseline and 3 h post fat load.

	Baseline	3 h	*p*-Value
Age (years)	43.4 ± 9.2	-	-
Sex (men/women)	13/26	-	-
BMI (kg/m^2^)	49.3 ± 7.2	-	-
Fasting glucose (mg/dL)	102.2 ± 14.5	-	-
HOMA-IR	5 ± 2.5	-	-
HbA1c (%)	5.7 ± 0.3%	-	-
TC (mg/dl)	192.6 ± 37.8	-	-
HDL-C (mg/dL)	44.9 ± 8.8	45.4 ± 10.4	0.666
LDL-C (mg/dL)	120.4 ± 31.7	121 ± 30.8	0.162
TG (mg/dL)	134.7 ± 59.2	206.1 ± 69.8	<0.001
Total Apo-B (mg/dL)	56.9 ± 32.5	61.3 ± 33.5	0.210
Apo-CIII (mg/dL)	19.6 ± 8	20.5 ± 7.1	0.300
Apo-AI (mg/dL)	153.3 ± 79.4	165.8 ± 95.6	0.020
PCSK9 (ng/mL)	187.2 ± 76.1	165.3 ± 70.8	<0.001
Zonulin (ng/mL)	545.3 ± 122.9	621.3 ± 278.6	0.040
LPS (EU/mL)	0.425 ± 0.007	0.433 ± 0.007	<0.001
LBP (µg/mL)	12.9 ± 0.8	13.3 ± 0.5	<0.001

TG: triglycerides; LDL-C: low-density lipoprotein cholesterol; Apo-B: apolipoprotein B; Apo-AI: Apolipoprotein AI; Apo-CIII: apolipoprotein CIII; LPS: lipopolysaccharide; LBP: lipopolysaccharide-binding protein; PCSK9: Proprotein convertase subtilisin kexin type 9.

**Table 2 biomolecules-10-00748-t002:** Correlation coefficients among baseline and 3 h post fat load PCSK9, clinical and anthropometrical characteristics, lipid parameters, and intestinal permeability.

	PCSK9 Baseline	PCSK9 3 h
Age (years)	0.133	0.247
*Antropometrics*		
BMI (kg/m^2^)	0.027	0.050
WC (cm)	−0.058	−0.045
*Insulin resistance*		
Glucose (mg/dL)	0.050	0.011
HOMA-IR	−0.011	−0.095
*Plasma lipids*		
TC baseline (mg/dL)	0.484 **	-
TC 3 h (mg/dL)	-	0.466 **
HDL-C baseline (mg/dL)	0.398 *	-
HDL-C 3 h (mg/dL)	-	0.478 *
LDL-C (mg/dL)	0.459 **	-
LDL-C 3 h (mg/dL)	-	0.411 *
TG baseline (mg/dL)	−0.014	-
TG 3 h (mg/dL)	-	−0.057
Total Apo-B baseline (mg/dL)	0.219	-
Total Apo-B 3 h (mg/dL)	-	0.246
Apo-CIII baseline (mg/dL)	0.262	-
Apo-CIII 3 h (mg/dL)	-	0.062
Apo-AI baseline (mg/dL)	0.058	-
Apo-AI 3 h (mg/dL)	-	0.009
*Intestinal permeability*		
Zonulin baseline (ng/mL)	0.184	-
Zonulin 3 h (ng/mL)	-	0.480 *
*Endotoxemia*		
LPS baseline (EU/mL)	−0.004	-
LPS 3 h (EU/mL)	-	−0.067
LBP baseline (µg/mL)	0.038	-
LBP 3 h (µg/mL)	-	−0.009

3 h: 3 h post fat load. * *p* < 0.05; ** *p* < 0.001. BMI: body mass index; WC: waist circumference; HOMA-IR, homeostatic model assessment of insulin resistance; TC: total cholesterol; LDL-C: low-density lipoprotein cholesterol; HDL-C: high-density lipoprotein cholesterol; VLDL-C: very low- density lipoprotein cholesterol; TG: triglycerides; Apo-B: apolipoprotein B; Apo-AI: Apolipoprotein AI; Apo-CIII: apolipoprotein CIII; LPS: lipopolysaccharide; LBP: lipopolysaccharide-binding protein; PCSK9: Proprotein convertase subtilisin kexin type 9.

**Table 3 biomolecules-10-00748-t003:** Multiple linear regression analysis for baseline PCSK9.

		PCKS9 Baseline	
	β	CI 95%	*p*-Value
Age (years)	0.027	−3.012–3.352	0.873
Sex	−0.299	−96.789–2.010	0.083
LDL-C (mg/dL)	0.370	0.014–1.942	0.030
TG baseline (mg/dL)	−0.007	−0.664–0.510	0.964
Zonulin baseline (ng/mL)	0.052	−0.172–0.448	0.760

Adjusted R-squared: 0.194; LDL-C: low-density lipoprotein cholesterol; TG: triglycerides; PCSK−9: Proprotein convertase subtilisin kexin type 9.

**Table 4 biomolecules-10-00748-t004:** Multiple linear regression analysis for PCSK9 3 h.

		PCKS9 3 h	
	β	CI 95%	*p*-Value
Age (years)	0.073	−1.557–3.668	0.624
Sex	−0.249	−15.543–100.035	0.073
LDL-C 3 h (mg/dL)	0.330	0.031–1.807	0.043
TG 3 h (mg/dL)	0.110	−0.199–0.438	0.441
Zonulin 3 h (ng/mL)	0.328	−0.193–0.149	0.035

Adjusted R-squared: 0.345; 3 h: 3 h post fat load. LDL-C: low-density lipoprotein cholesterol; TG: triglycerides; PCSK9: Proprotein convertase subtilisin kexin type 9.

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
