# Peer review of "Relationship of Zonulin with Serum PCSK9 Levels after a High Fat Load in a Population of Obese Subjects"

_biomolecules, 2020, doi:10.3390/biom10050748_

Round 1
Reviewer 1 Report
The authors answered the main questions and accepted some modifications.
However, I think that the authors should improve the limitations, namely to put: "this study was designed in order to confirm an exploratory hypothesis and a control group was not included. Probably in the future, we will prepare a new study with a control group."
Author Response
Reviewer 1
The authors answered the main questions and accepted some modifications.
Comment 1: However, I think that the authors should improve the limitations, namely to put: "this study was designed in order to confirm an exploratory hypothesis and a control group was not included. Probably in the future, we will prepare a new study with a control group."
Response: First of all, we thank the reviewer for his/her comments.
We have added your suggestion in the limitations of the study.

Reviewer 2 Report
The authors have implemented most comments of the reviewers to improve quality of the manuscript.
Yet one critical point remains which has to be clearly stated, specified and discussed:
The authors have determined serum LPS and LBP to strengthen the link to intestinal permeability. The measured increase in LPS is rather small (this is in line with previous reported data from the authors in Clemente-Postigo et al. JLipidRes 2012) and no correlation was found with PCSK9 levels. Thus, while increased surrogate markers of intestinal permeability increase in parallel to a decrease in PCSK9, the only link to intestinal permeability remain the finding of an independent relation of 3h PCSK9 levels with 3h zonulin levels.
As already indicated by the previous reviewer, measuring zonulin using the most frequently used ELISA kits from IDK Immundiagnostik (Germany) or Cusabio (China) is not only inaccurate, it is inadequate for measuring zonulin (identified and defined as prehaptoglobin2). Two recent publications have independently demonstrated that commercially available ELISA kits do not detect prehaptoglobin 2, but rather unspecifically measure other proteins potentially related to the ‘zonulin family’ (Scheffler et al. Front Endocrinol 2018; Ajamian et al. PONE 2019). Furthermore, the roles in gut leakiness and bacterial translocation of the proteins measured by these ELISA kits remain unclear. Thus, until validated ELISA are available, it is important for the scientific and medical community to exercise caution when considering the measurement of serum zonulin as a biomarker of intestinal permeability.
Also the Gut paper (Gut. 2020 Jan;69(1):191-193. doi: 10.1136/gutjnl-2018-317726) referenced by the authors in the rebuttal has used the ELISA from Cusabio and thus is inadequate to advocate and justify measuring zonulin as a biomarker for intestinal permeability.
In conclusion, the authors, using a zonulin ELISA, clearly detect a relation of PCSK9 levels with a factor/factors that have been previously shown to be significantly correlated with intestinal permeability, inflammation and insulin resistance. Yet, taking into account the evidence mentioned above, it is incorrect to state that this factor is zonulin. This has to be discussed and clearly stated in the manuscript.
Thus, before acceptance for publication, the authors have to make the following amendments:
1) Please give exact information, which zonulin ELISA was used, as MyBioSource offers multiple zonulin ELISA. Most of them are either identical to the IDK Immundiagnostic or Cusabio ELISA and thus are not measuring zonulin=prehaptoglobin2.
2) The authors must clearly state and discuss the limitation of measuring “zonulin” or zonulin-like or zonulin-family-protein as a biomarker for intestinal permeability using ELISA.
3) As the relation of “zonulin” levels is the only and weak link between PCSK9 and intestinal permeability, the authors have to down-tune the potential relationship of PCSK9 with intestinal permeability, also in the title.
Author Response
Reviewer 2
Comment 1: The authors have implemented most comments of the reviewers to improve quality of the manuscript.
Yet one critical point remains which has to be clearly stated, specified and discussed:
The authors have determined serum LPS and LBP to strengthen the link to intestinal permeability. The measured increase in LPS is rather small (this is in line with previous reported data from the authors in Clemente-Postigo et al. JLipidRes 2012) and no correlation was found with PCSK9 levels. Thus, while increased surrogate markers of intestinal permeability increase in parallel to a decrease in PCSK9, the only link to intestinal permeability remain the finding of an independent relation of 3h PCSK9 levels with 3h zonulin levels.
As already indicated by the previous reviewer, measuring zonulin using the most frequently used ELISA kits from IDK Immundiagnostik (Germany) or Cusabio (China) is not only inaccurate, it is inadequate for measuring zonulin (identified and defined as prehaptoglobin2). Two recent publications have independently demonstrated that commercially available ELISA kits do not detect prehaptoglobin 2, but rather unspecifically measure other proteins potentially related to the ‘zonulin family’ (Scheffler et al. Front Endocrinol 2018; Ajamian et al. PONE 2019). Furthermore, the roles in gut leakiness and bacterial translocation of the proteins measured by these ELISA kits remain unclear. Thus, until validated ELISA are available, it is important for the scientific and medical community to exercise caution when considering the measurement of serum zonulin as a biomarker of intestinal permeability.
Also the Gut paper (Gut. 2020 Jan;69(1):191-193. doi: 10.1136/gutjnl-2018-317726) referenced by the authors in the rebuttal has used the ELISA from Cusabio and thus is inadequate to advocate and justify measuring zonulin as a biomarker for intestinal permeability.
In conclusion, the authors, using a zonulin ELISA, clearly detect a relation of PCSK9 levels with a factor/factors that have been previously shown to be significantly correlated with intestinal permeability, inflammation and insulin resistance. Yet, taking into account the evidence mentioned above, it is incorrect to state that this factor is zonulin. This has to be discussed and clearly stated in the manuscript.
Response: Again, we thank the reviewer for the time spent reviewing our article and the constructive criticism he/she has offered us.
Thus, before acceptance for publication, the authors have to make the following amendments:
Comment 2: 1) Please give exact information, which zonulin ELISA was used, as MyBioSource offers multiple zonulin ELISA. Most of them are either identical to the IDK Immundiagnostic or Cusabio ELISA and thus are not measuring zonulin=prehaptoglobin2.
Response: We have specified the catalog number of the MyBioSource zonulin ELISA kit used: MBS749365
Comment 3: 2) The authors must clearly state and discuss the limitation of measuring “zonulin” or zonulin-like or zonulin-family-protein as a biomarker for intestinal permeability using ELISA.
Response: We thank the reviewer for his/her pertinent commentary. We have added some sentences clarifying the limitations of the zonulin kit used.
Comment 4: 3) As the relation of “zonulin” levels is the only and weak link between PCSK9 and intestinal permeability, the authors have to down-tune the potential relationship of PCSK9 with intestinal permeability, also in the title.
Response: Following the reviewer´s recommendation, we have modified the title of our manuscript.

This manuscript is a resubmission of an earlier submission. The following is a list of the peer review reports and author responses from that submission.
Round 1
Reviewer 1 Report
Summary:
The authors conducted a fat-tolerance test on 39 morbidly obese patients awaiting bariatric surgery. The authors measured plasma TG, LDL-C, Total apoB, ApoCIII, Apo A1, Pcsk-9, and Zonulin at baseline and at 3 hours post the fat meal. The authors describe the main finding to be that plasma Pcsk-9 decreased significantly post-prandially and in turn was correlated with plasma zonulin.
Main Comments:
The authors present an intriguing hypothesis that PCSK-9 expression may influence intestinal permeability and be involved with extravasation of bacterial products that contribute to a systemic proinflammatory response following a fatty meal. While the reviewer finds this an exciting and plausible area of research, the interpretation of the current studies are not necessarily supported by the data collected. Some examples are provided for consideration:
Main conclusion seems to be confusing:
Is it “Therefore the pcsk9 reduction after fat load could help clear the higher plasma concentrations of LPS produced by the increase of intestinal permeability as a consequence of high meal in obese subjects”
Or is it “…suggesting the possibility of fat load on pcsk9 may be determined by the changes induced by fat over gut microbiota and intestinal permeability”
The link between intestinal permeability is quite weak. The manner in which zonlulin is measured in plasma is confounded and may not represent intestinal-specific response (it may also include zonulin from the liver?). Other markers of intestinal permeability should be measured in order to strengthen this relationship (recent literature suggests that current methods to measure Zonulin are not accurate https://www.ncbi.nlm.nih.gov/pmc/articles/PMC6331146/)
Please make the conclusion more specific. The authors could perhaps measure LPS, as Zonulin is a surrogate marker of permeability. Additionally measurement of LPS will help with the final conclusion.
In the multiple linear regression in table 3 and 4, the female sex is not included. Please have a separate table/model with female sex. You can present multiple models with additional variables. An example can be found here:
https://journals.plos.org/plosone/article/file?id=10.1371/journal.pone.0114281&type=printable
Similarly apoA1 was significantly different post prandially and is not included in the multiple linear regression.
Please improve the language used for the manuscript throughout, as the understandability of the written word is unclear in some sections.
Minor comments:
Please include a table of all the baseline characteristics separately.
In table 2, the variables can be categorized as Anthropometrics, Insulin resistance, and plasma lipids.
For table 3 and 4 use a standard table format as is routinely reported in literature for multiple linear regression. It should include, SE, Std coefficient, Percent of variance explained etc.
Reviewer 2 Report
Molina-Vega and colleagues present a hypothesis that PCSK9 levels may be related to intestinal permeability.
In my opinion, the study presents a good background and fundament, the sample being a little small, but with an acceptable number. The authors discuss this point in the limitations.
In my opinion the results are confusing and unclear. I do not understand table 2 and suggest that the anthropometrical and clinical characteristics are shown in Table 1.
would it be possible to put some graphic? in my opinion it would become more perceptible.
be aware that whenever you refer to a gene you must write it in italics.
in my opinion the article would improve if they had a control group of people with normal weight to the same feeding conditions and not exercise.